# Impact of Violent Experiences and Social Support on R-NSSI Behavior among Middle School Students in China

**DOI:** 10.3390/ijerph18073347

**Published:** 2021-03-24

**Authors:** Kun Liu, Xueyan Yang, Moye Xin

**Affiliations:** 1Institute for Population and Development Studies, School of Public Policy and Administration, Xi’an Jiaotong University, Xi’an 710049, China; lk0536@163.com (K.L.); xjxj4133@163.com (M.X.); 2School of Public Health and Management, Binzhou Medical University, Yantai 246003, China

**Keywords:** middle school students, R-NSSI, social support, violent experiences, sex differences, China

## Abstract

Repetitive nonsuicidal self-injury (R-NSSI) is an extreme manifestation of nonsuicidal self-injury (NSSI) behavior that causes bodily harm and emotional and personality disorders. It is a growing concern, especially among adolescents; therefore, this study aims to provide empirical support for effective interventions on R-NSSI behavior among adolescents in China. We used data of about 1180 students from a survey conducted in seven middle schools in Xi’an, China, and applied multiple logistic regression to analyze NSSI and R-NSSI among male and female students, including their influencing factors. We found no significant difference between male and female students’ R-NSSI; however, regarding influencing factors, male students had more violent experiences and less social support than female students. Parental and familial factors played the most prominent role in social support. Social support was found to be a main-effect mechanism in the effect of violent experiences on R-NSSI among male students, whereas the mechanism had both a main effect and a certain buffer effect among female students. R-NSSI was found to be more prevalent among younger children, children with siblings, and those with romantic relationship experiences. We also found that healthy adolescent development involves the participation of families and schools. Health education should be conducted according to the students’ sex and characteristics.

## 1. Introduction

Repetitive nonsuicidal self-injury (R-NSSI) behavior refers to the intentional and repeated form of nonsuicidal self-injury (NSSI) behavior, that is, the repeated harming of one’s body without committing suicide. Moreover, R-NSSI is an “addiction”—it is the extreme behavior of causing destructive harm to the body and emotional and personality disorders, like psychological problems, leading to heightened pain tolerance, reduced fear of death, and increased risk of suicide. 

Adolescents have a high incidence rate for NSSI; half of those who experience NSSI also experience R-NSSI [1]. The tendency of R-NSSI among adolescents often means that they have long-term or periodic psychological problems. It is even believed that R-NSSI is caused by multiple disease entities [2]; therefore, it poses a more significant threat to adolescents’ healthy development. NSSI is a significant concern in most cultures. Under the contemporary Chinese education system, the middle school stage (11–16 years) is a highly valued period. The body and mind of middle school students undergo significant changes, and they are particularly susceptible to the influence of the external environment. Chinese middle school students experience significant pressure due to school examinations and admission. Therefore, the incidence of NSSI among adolescents is high and is a cause for great concern. Adolescents with R-NSSI are more likely to use avoidance or venting as coping strategies and to report multiple types of trauma, drug and alcohol abuse, suicidal ideation, and more organic injuries than adolescents with one-time NSSI [3]. Therefore, R-NSSI has more serious behavioral consequences and more complex influence mechanisms. Consequently, it is important to study the influence mechanisms of NSSI, especially R-NSSI, and adopt timely, effective intervention measures to help the physical and mental health of adolescents. This study analyzes the influence mechanisms of R-NSSI behavior using the factors of early violent experiences and current social support of adolescents.

According to the empirical avoidance model, R-NSSI is a group of behaviors appertaining to the desire to avoid inner experiences, such as anxiety and depressed thoughts, after experiencing physical, sexual, or even cold violence (an emotional form of violence characterized by a complete withdrawal of all physical, financial, spoken, and emotional communication). In particular, child abuse not only harms the body, but it also causes psychological problems, such as inferiority complex, depression, and personality disorder, leading to a higher incidence of suicidal ideation and behavior than among adolescents who have not been abused [4]. Research shows that up to 79% of self-injurers have experienced child abuse or bullying [5]. Most theoretical models also consider that early adverse experiences and negative sexual events have an important impact on R-NSSI behavior among adolescents [6,7,8]. International studies show that 20% to 30% of students have been subjected to verbal and visual violence, with verbal violence being a common factor in all kinds of NSSI, leading them to become both victims and perpetrators [9]. In summary, negative sexual life events, such as violence, abuse, and bullying, cause physical and mental trauma in adolescents and have a direct or indirect impact on their R-NSSI behavior. 

The crucial protective role of social support has been proven in many studies. For example, Muehlenkamp [10] found that perceived social support with R-NSSI was significantly lower than that for the control participants with and without NSSI. Additionally, people with R-NSSI believe that talking to others about NSSI is unhelpful; thus, they are less likely to seek help or advice from others. This indicates that social support has a significant direct impact on depressive stress and post-trauma individual development [11]. In this regard, a series of important social support dimensions for students, which include family, school, and peer support, have been thoroughly explored in China and worldwide. First, the self-determination theory holds that teacher support for students’ basic psychological needs is an important promoting factor within the school environment and the overall atmosphere, while perceived support from teachers is a protective factor for the psychological needs of adolescent students with NSSI behavior [12]. Second, familial support is the most important predictor and intervening factor for R-NSSI among adolescents. A good quality relationship between adolescents and their parents can prevent NSSI among adolescents, showing its protective mechanism [13]. Adolescents have less NSSI behavior when parents support their decisions and maintain open communication, which also leads to better recovery from trauma injuries and improved parental attitudes and cooperation regarding school. Conversely, life events, such as tension with family members, a conflict between mother and child, and lack of warmth from parents [14], have become important reasons for R-NSSI among adolescents. Third, adolescent attitudes and behaviors are strongly influenced by peer group norms and relations. Bad friendships, hostility, and violent peer environments increase the occurrence of NSSI in students [15], while peer groups with high impulsive responses to negative emotions might increase the risk of NSSI among its members [16]. These all reflect the influence of the important dimensions of social support on R-NSSI behavior among adolescents.

Problem behavior theory is a systematic, multivariable, and comprehensive theoretical system of social psychology. Jessor [17], in his longitudinal tracking research of high school and college students in the 1990s, considered individual problem behavior a type of learned functional behavior, resulting from the interaction between the personality system and the perceived social environment system. This theory is integrated into the basic idea of the two-factor theory [18], which holds that the occurrence of problem behaviors has corresponding risk and protective factors that form a comprehensive theoretical model. This theory suggests that family, company, school, and community, as well as risk and protective factors, directly impact problem behavior. At the same time, protective factors are likely to affect risk factors and the relationship between problem behavior regulation, thereby weakening the influence of risk factors on problem behavior. In other words, protective factors stimulate prosocial behavior and improve the individuals’ ability for psychosocial rehabilitation. Hence, previous studies have proposed the hypothesis of the main-effect and buffer-effect models based on the mechanism of social support concerning health-related issues. On the one hand, according to the main-effect model, the promoting effect of social support is generalized—the more support there is, the more positive the effect is [19]. On the other hand, according to the buffer model, individual health-related behaviors are affected by the situation, and social support plays the role of adjustment and buffer in this process [20]. Meanwhile, other scholars believe social support is the main effect that directly affects individuals’ health-related behaviors and acts as a buffer in the given situation [21].

Many previous studies have provided important references for the discussion of R-NSSI among adolescents, but there is more work to be done. First, the existing research focuses on R-NSSI behavior as an extreme manifestation of NSSI. Thus, the consequence is more serious, which prompts the question: are there any special patterns and characteristics in R-NSSI behavior among Chinese male and female middle school students? Second, few existing studies consider the relationship of both violent experiences and social support with R-NSSI. According to the problem behavior theory, whether the protective factor of social support acts as the main effect or the buffer effect in its influence on the risk factors of violent experiences regarding R-NSSI needs to be verified. Therefore, applying the basic theoretical framework of problem behavior theory (Figure 1) from a sex perspective, this study aims to build an R-NSSI influence factor analysis model for adolescents. Ultimately, the objective of this study is to explain its behavior mechanism and provide empirical support for effective interventions on R-NSSI behavior among male and female middle school students in China from the perspective of violent experiences and social support.

## 2. Materials and Methods 

### 2.1. Data

This study was based on the “Health Risk Behavior Survey of Middle School Students” in 2015, conducted on middle schools in Shaanxi province, China. The stratified proportion sampling method was adopted for recruiting boys and girls from Grades 7 to 12 of seven middle schools in Xi’an (four key schools and three ordinary middle schools). The seven middle schools were The High School Affiliated To Shaanxi Normal University, The Middle School Attached To Northwestern Polytechnical University, Xidian High School Attached To Xidian University, The High School Affiliated To Xi’an Jiaotong University, and Xi’an Tie Yi High School, Taiyi Road Middle School, and Qinchuan Middle School. In cooperation with each school’s liaison teachers, students from different grades and sexes were selected in equal proportion. Out of 1200 questionnaires, 1180 valid questionnaires were returned by 655 male and 525 female students, with a response rate of 98.33%. 

### 2.2. Variable Measurement 

The study involved three core variables: R-NSSI among adolescents, violent experience, and social support.

There is no clear and universally applicable definition of R-NSSI, and the number of R-NSSI incidents is not the same. Based on the study of Howe-Martin [22], we classified NSSI into “N-NSSI” (non-NSSI), “O-NSSI” (one-time NSSI), and “R-NSSI” (repetitive NSSI) by asking participants the following indefinite question: “Have you ever taken the following nonsuicidal actions to intentionally hurt yourself?” The answers were rated from 0 (*none of the above*) to 2 (*selected two or more options*). The dependent variables were defined as multicategorical variables.

According to the World Health Organization and correlational research [23], violent experience refers to injuries, violence, and bullying and is classified into five specific types: verbal violence, physical violence, visual violence, cold violence, and sexual violence. Participants answered the question “have you ever experienced verbal violence/physical violence/visual violence/cold violence/sexual violence?” using a 5-point Likert scale ranging from 1 (*never*) to 5 (*always*). We then calculated the average score of each violent experience and refined the content by setting the perpetrators of various violent experiences and dating the encounter to help explain and analyze the results.

We used Wang’s Perceived Social Support Scale for students [24], which is widely used in research and has good reliability and validity. Participants answered 12 questions that are divided equally into three dimensions (family support, friend support, and other support) using a 5-point Likert scale ranging from 1 (*totally agree*) to 5 (*totally disagree*). A higher score means higher support in the specific dimension. Personal and family information were control variables, age was a continuous variable, being an only child and romantic relationship experience were binary variables, and parents’ feelings and family’s financial status were ordinal variables.

### 2.3. Data Analysis

SPSS Statistics version 22.0 software (SPSS Inc., Chicago, IL, USA) was used for processing and analyzing data. We calculated and compared the scores of violent experience and social support between boys and girls and tested the other control variables for differences. Then, we performed multiple logistic regression analyses for boys and girls using violent experience, social support, and the control variables. Using students without NSSI as the reference, the influencing factors of primary NSSI and R-NSSI were analyzed.

## 3. Results

The participants included 655 male and 525 female students, with a male-to-female ratio of 1.25 to 1 and an average age of 14.63 years. Among them, 648 were only-children, accounting for more than half of all middle school students, with 374 male and 274 female students accounting for 57.7% and 52.3% of the total male and female participants in this group, respectively. The proportion of only-sons was slightly higher than that of only-daughters (X^2^ = 3.451, *p* < 0.1). Moreover, there was no significant difference between sexes among 311 middle school students in terms of romantic relationship experiences, of which 184 were male and 127 were female, accounting for 29.1% and 25.0% of all male and female middle school students, respectively. 

The main variables of the respondents are shown in Table 1. The participants’ average NSSI rate was 24.05%, accounting for 13.1% and 15.6% of all male and female middle school students, respectively. The chi-square test showed no significant difference between middle school students by sex. Here, we address the first problem listed above. In terms of violent experience, the scores for verbal violence (F = 5.347, *p* < 0.001), physical violence (F = 4.406, *p* < 0.001), and visual violence (F = 12.219, *p* < 0.001) of male middle school students were significantly higher than those of female students, while there was no significant difference in cold violence and sexual violence. In terms of social support, male students scored lower than female students in all three dimensions of family support (F = −2.900, *p* < 0.01), friend support (F = −1.803, *p* < 0.1), and other support (F = −3.639, *p* < 0.001). The details are shown in Table 1. 

The regression analysis on the influencing factors of primary NSSI and R-NSSI among male middle school students is shown in Table 2. 

Compared with those classified under non-NSSI, the probability of one-time NSSI increased 1.704 times with each increase in the frequency of sexual violence (Model 1), which was significant (1.704, *p* < 0.05). Moreover, when the social support variable was added, the influence of the original violent experience disappeared; the probability of one-time NSSI increased 1.129 times with each increase in the frequency of cold violence (Model 2), which was significant (1.129, *p* < 0.01). 

However, when the variables of age, being an only child, experience in a romantic relationship, parents’ relationship status, and family’s financial status were added as control variables, there was a negative effect (0.530, *p* < 0.1) concerning one-time NSSI behavior (Model 3). Moreover, when the social support variable was added, the one-time NSSI behavior probability decreased by 0.530 times. Furthermore, it can be seen from Model 3 that age had a significant negative impact on the occurrence of one-time NSSI among male students (0.627, *p* < 0.01); that is, with an increase in age, the probability of one-time NSSI in male students decreased by 0.627 times. In other words, the younger the male students were, the more likely they were to have engaged in one-time NSSI.

Moreover, physical violence (1.787, *p* < 0.001), visual violence (1.267, *p* < 0.05), cold violence (1.662, *p* < 0.001), and sexual violence (1.993, *p* < 0.001) had significant positive effects on R-NSSI. This indicated that a more violent experience raised the likelihood of R-NSSI among male middle school students. However, there were no significant changes even after adding the social support variable, while family support had a significantly negative influence (0.491, *p* < 0.01) on male students’ R-NSSI behavior (Model 2). This indicated that family support lessened the effect mechanism of self-injury and reduced the incidence of R-NSSI by 0.491 times. In other words, greater familial help and support led to a lesser likelihood of R-NSSI among male middle school students. 

However, when the variables of age, only-child status, parents’ relationship status, family’s financial status, and experience in a romantic relationship were added as control variables, we found that visual violence lessened R-NSSI among male middle school students, while there were no obvious changes for cold violence, physical violence, and sexual violence. Similarly, family support had a significant negative influence on R-NSSI behavior, indicating that help and support from the family could lessen the occurrence of R-NSSI behavior among male middle school students. Moreover, age had a significant negative impact on R-NSSI among male middle school students (0.634, *p* < 0.001), while being an only child had a negative correlation (0.562, *p* < 0.1). Meanwhile, experience in a romantic relationship had a significantly positive correlation (2.398, *p* < 0.05). In other words, the older the male students were, the less likely they were to have engaged in R-NSSI; an only child was less likely to have engaged in NSSI than a child with siblings; those who had experienced a romantic relationship were more likely to engage in R-NSSI than those that had not. Above all, it can answer the second problem about male students; the main effect of social support on R-NSSI among male middle school students was verified but not the buffer effect. 

With the gradual addition of variables from Models 1 to 3, the explanatory power of the model was gradually strengthened. Cox–Snell *R*^2^ and Nagelkerke *R*^2^ in Model 1 were 0.14 and 0.189, respectively, while these values increased to 0.217 and 0.294, respectively, in Model 2, indicating that social support has a strong explanatory power concerning NSSI behavior among male middle school students. In Model 3, Cox–Snell *R*^2^ and Nagelkerke *R*^2^ further increased to 0.265 and 0.358, respectively, indicating that violent experience and social support had strong explanatory power with regard to NSSI behavior among male middle school students.

The regression analysis on the influencing factors of primary NSSI and R-NSSI among female middle school students is shown in Table 3.

Compared with those classified under non-NSSI, the probability of NSSI increased 0.787 times with each increase in the frequency of sexual violence (Model 4), which was positively significant (1.787, *p* < 0.05). Moreover, when the social support variable was added, the positive effect of sexual violence disappeared, indicating that all kinds of care and help could reduce the trauma caused by sexual violence against female middle school students (Model 5). 

Furthermore, visual violence had a significant positive influence on age (1.693, *p* < 0.05), while being an only child had varying degrees of negative effects on self-injury behavior (0.623, *p* < 0.001; 0.45, *p* < 0.1), indicating that older female middle school students were less likely to engage in NSSI than younger ones, and only-children were less likely to engage in NSSI than those with siblings (Model 6). 

Compared with those classified under non-NSSI, the probability of R-NSSI increased 1.655 times with each increase in the frequency of visual violence (Model 4), which was positively significant (1.655, *p* < 0.001). Moreover, after adding the social support variable, visual violence experience had a significantly positive effect (1.505, *p* < 0.05) on female students’ R-NSSI behavior, but the coefficient was slightly lower (Model 5). This showed that social support had a certain regulatory effect, while verbal violence experience had a significant positive effect on R-NSSI among female students (1.510, *p* < 0.05). In addition, social support could strengthen the effects of violence on R-NSSI to a certain extent among female middle school students. 

However, the newly added family support variable had a significant negative effect on R-NSSI among female middle school students (0.578, *p* < 0.05), reflecting its main effect. This indicated that every increase in the degree of support and help from the family led to a decrease of 0.578 times in the probability of R-NSSI among female middle school students. Additionally, there were no significant changes in the direction, size, and significance of verbal violence, visual violence, and family support even after controlling for variables like age, only-child status, experience in a romantic relationship, parents’ emotional status, and family financial status (Model 6). Moreover, age had a negative crude correlation with R-NSSI among female middle school students (0.813, *p* < 0.1), indicating that each yearly increase in the age of female middle school students decreased the probability of R-NSSI by 0.813 times. Simultaneously, parents’ feelings had a significant negative impact on female middle school students’ R-NSSI (0.733, *p* < 0.05), indicating that each increase in parents’ feelings decreased the probability of R-NSSI by 0.733 times. Above all, it also can answer the second problem of female students; we verified the main effect and a certain buffer effect of social support on R-NSSI among female middle school students. 

With the gradual addition of variables from Models 4 to 6, the explanatory power was gradually strengthened. In Model 4, Cox–Snell *R*^2^ and Nagelkerke *R*^2^ were 0.96 and 0.123, respectively, which increased to 0.129 and 0.166, respectively, in Model 5. This showed that social support had strong explanatory power with regard to NSSI behavior among female middle school students, more so after including the control variables in Model 6. Cox–Snell *R*^2^ and Nagelkerke *R*^2^ further increased to 0.228 and 0.292, respectively, showing that violent experience and social support have strong explanatory power with regard to female middle school students’ self-injury behavior.

## 4. Discussion

There are limitations in the definition and measurement of R-NSSI standards in the current Chinese and international literature. There is no clear, universally applicable, and accurate definition of R-NSSI; its time range and frequency have been defined differently in some studies [25], and its horizontal comparison rate lacks an exact basis. In studies conducted outside China, NSSI is sometimes regarded as a symptom of mental illness, while some consider it an independent disease entity [26]. Other studies have combined psychopathological factors and directly taken clinical samples or hospitalized patients as research participants [27]. Therefore, more occurrences of R-NSSI were identified in those studies. A few studies have focused on R-NSSI in China, and, considering the great difference between the general population and the clinical samples of those with mental illness, R-NSSI was still identified more than once among adolescents in this study. By studying the influence mechanism of R-NSSI among middle school students, the following problems were identified: (1) The influence of a violent experience on R-NSSI behavior among middle school students differs according to sex; (2) social support plays different roles in the influence of a violent experience on R-NSSI behavior according to sex; (3) other factors influence students’ R-NSSI behavior, with different effects.

### 4.1. The Influence of Violent Experiences on R-NSSI Behavior among Middle School Students by Sex

With regard to sex differences in NSSI behavior, there was no significant difference in the incidence of R-NSSI behavior between male and female middle school students in China, but there were significant differences in the influencing factors. Previous studies have suggested that NSSI has a higher incidence among women [28], while others have found a higher incidence among men [29]; these results support the view that sex differences may not be universal [30]. They aligned with our finding of different scores for the experience of violence for male and female middle school students. Our results showed that the R-NSSI rates of male and female middle school students were 13.1% and 15.6%, respectively, which are similar to the results of the community survey conducted by Howe-Martin [22]. Male students reported significantly higher scores in verbal violence, physical violence, and visual violence and significantly lower scores in all three dimensions of social support than female students. This shows that male students who suffer more serious violence receive less social support than female students. Further regression analysis confirmed that although the results of the survey among male and female students in NSSI were similar, the internal factors were different. All kinds of violent experiences differed for R-NSSI behavior among male and female students—physical, visual, cold, and sexual violence had significant effects on male students’ R-NSSI, while visual violence had a significant effect on female students’ R-NSSI. In general, there were certain sex differences in the influencing factors of R-NSSI among middle school students. Therefore, the sex characteristics of middle school students cannot be ignored in education, and different intervention methods should be adopted accordingly. At the same time, the results suggest that care and protection should be given to both boys and girls. As some aspects of injuries suffered are more hidden, more attention should be given to the affected adolescents.

### 4.2. The Role of Social Support in the Influence of Violent Experiences on R-NSSI Behavior by Sex 

We verified the main effect of family support on the influence of violent experiences on R-NSSI and its moderating effect among female middle school students. We found that family support had a positive effect in reducing R-NSSI behavior among male and female middle school students—the higher the degree of family support, the less likely it was for male and female middle school students to have R-NSSI, which verified the main effect mechanism of social support. For female middle school students, the addition of social support strengthened the significance of verbal violence and weakened the significance of visual violence. This indicates that social support plays a moderating role in the influence of verbal and visual violence on R-NSSI behavior among female middle school students. 

Moreover, after adding the control variables, we found that parents’ feelings had a prominent influence, indicating female middle school students were more dependent and affected by their families, especially their parents [31]. Additionally, both male and female middle school students acknowledged the importance of parenting and having a positive family environment to protect adolescents against certain negative psychological and behavioral patterns [32,33]. Effective parenting and having a positive family environment can help students leave the shadow of violence and reduce the reoccurrence of NSSI behavior. Conversely, ineffective parenting and a negative family environment could increase the violence experienced by middle school students, which would increase R-NSSI risk, especially among women. Nevertheless, according to relevant studies on family rearing, it has been proven that warm and receptive parenting is most worthy of being advocated, while overregulated parenting, characterized by coldness and rejection, is bound to backfire [34].

### 4.3. Other Factors Influencing Students’ R-NSSI Behavior

Age can affect the probability of R-NSSI among male and female middle school students—the younger the middle school students were, the higher their probability of R-NSSI. This further confirms the results of relevant studies that have stated that early adolescents are more likely to release their emotions through negative coping methods than middle adolescents [35]. This is because the older the individuals become, the more mature their minds will be; hence, when facing various problems and difficulties in life, they will be more rational and take more appropriate steps in dealing with them. After puberty, an increase in age leads to a decrease in the probability of R-NSSI.

Moreover, NSSI incidence among those with siblings is higher than those who were only-children. Only-daughters had a lower probability of having O-NSSI than middle school students with siblings, while only-children have a lower probability of having R-NSSI. This finding was contrary to the conclusion of Liu [36] in college students but consistent with the meta-analysis results of scholars regarding NSSI among middle school students in mainland China [37]. This shows that the biggest advantage of only-children over those with siblings is that they receive more attention from their parents and families. Thus, when negative events or emotions appear, parents can deal with them promptly, which can effectively prevent extreme behaviors like NSSI. This also provided strong evidence to refute the social prejudice in China that only-children are a problem group [38]. In addition, it indirectly reflects the difference between middle school and college students. In future studies, we aim to compare other behavioral differences between middle school and college students based on whether they are an only child or not.

Finally, in this study, male middle school students who had experienced a romantic relationship were more likely to have R-NSSI. Middle school students are young, immature, and emotionally unstable. Especially in today’s traditional Chinese education environment, middle school students’ romantic relationships are generally regarded as “puppy love” and are often forbidden by their schools, teachers, and parents [39]. Under the pressure of strict control, some may feel the need to take great risk to hide their relationship. If the romantic relationship is unstable, other emotional crises are bound to increase their troubles. Studies have shown that the proportion of injuries related to romantic relationships among males is higher than that among females [40], which may be related to the fact that boys are not as good at expressing their troubles through communication as girls are. Therefore, male students are more likely to vent or solve their problems through NSSI, while those who have experienced romantic relationships are more likely to have R-NSSI.

### 4.4. Limitations and Prospects

This study has the following limitations. First, since data collection was conducted only in Xi’an, China, the sample has certain geographical limitations that cannot explain the overall situation of Chinese middle school students. Second, owing to the lack of a clear definition of R-NSSI in academia, there is no accurate measurement standard for horizontal comparison. In addition, this study considered only the influence of certain variables, namely, being an only child, experience in a romantic relationship, family’s financial status, and parents’ feelings on NSSI and R-NSSI. We did not consider other possible influencing factors, such as academic pressure, school health education, and parents’ and teachers’ education levels. In future studies, we seek to improve the representativeness of the sample by including participants from cities in East and Central China. Moreover, latent class analysis and other methods should be used to divide the population of R-NSSI effectively, and the frequency and degree of R-NSSI should be defined scientifically and measured effectively to provide an accurate basis for horizontal and vertical comparison. Furthermore, future research should comprehensively consider risk factors and protective factors related to the R-NSSI of middle school students in terms of personal, family, school, society, and other aspects to thoroughly explore the influencing factors and complex mechanisms of R-NSSI of middle school students.

## 5. Conclusions

This study focused on R-NSSI behavior among middle school students and explored the mechanisms of violent experiences and social support. We found there was little difference in R-NSSI incidence between male and female middle school students, but there were differences in the influencing factors. First, male students had a higher degree of violent experiences than female students, but support from their family, friends, and other aspects was lower. Second, among the dimensions of social support, factors related to parents and family played the most prominent role, being the main-effect mechanism in the influence of violent experience on R-NSSI behavior among male middle school students. However, for female middle school students, both the main effect and certain buffer effects were present. Thus, social support, as a protective factor, could have both a main effect and a buffer effect on the influence of the violent experience risk factor on R-NSSI among middle school students. Additionally, we found that age, being an only child, and experience in a romantic relationship had significant effects on R-NSSI among middle school students. Based on these findings, we suggest middle school is a crucial transition stage for minors. 

Middle school students are the future of family, society, and country, and their healthy development requires joint participation and maintenance by their family, schools, and society. Therefore, we suggest the following. First, from the perspective of sex, the “crisis” of boys’ NSSI behavior is deepening. Their vulnerable status is more concealed than that of girls, and they are often more easily ignored in China. Therefore, care and protection, whether at home, in school, or in society, should be aimed at both girls and boys. Second, parents are the first teachers of children. Thus, health education is the responsibility of the school and the family. Therefore, it is urgent to innovate, explore, and implement a new model of homeschool health education and create a good environment and atmosphere for the healthy development of adolescents. Finally, in the context of the “Healthy China 2030” strategy, individualized health education intervention plans should be formulated around the growth background and experience of middle school students and other personalized characteristics. Individualized health education intervention methods should be innovated to intervene comprehensively and effectively in the health risk behaviors of middle school students.

## Figures and Tables

**Figure 1 ijerph-18-03347-f001:**
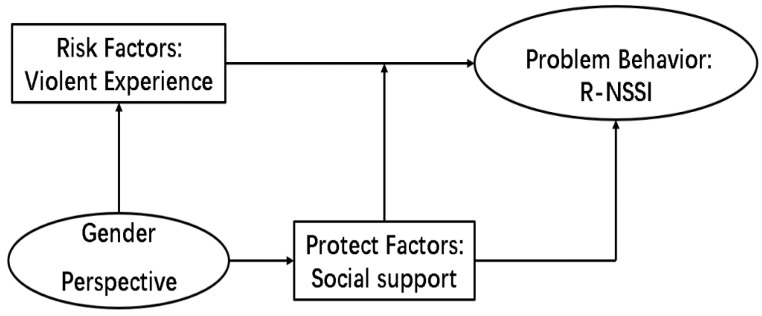
Problem behavior theory explanation model of repetitive nonsuicidal self-injury (R-NSSI) behavior of adolescents.

**Table 1 ijerph-18-03347-t001:** Sex differences in variable measurement.

Variables	Male (*N* = 655)	Female (*N* = 525)	X^2^/F
	N/M (PCT/SD)	N/M (PCT/SD)	
Types of NSSI			2.714
N-NSSI	511 (78%)	388 (73.9%)	
O-NSSI	58 (8.9%)	55 (10.5%)	
R-NSSI	86 (13.1%)	82 (15.6%)	
Violent experience			
verbal violence	2.390 (±1.077)	2.064 (±0.973)	5.347 ***
physical violence	1.962 (±0.893)	1.731 (±0.851)	4.406 ***
visual violence	2.673 (±1.276)	1.852 (±0.894)	12.219 ***
cold violence	1.904 (±1.020)	1.966 (±0.912)	−1.063
sexual violence	1.152 (±0.614)	1.068 (±0.433)	2.540
Social support			
family support	3.350 (±1.088)	3.534 (±1.010)	−2.900 **
friend support	3.447 (±1.099)	3.560 (±0.988)	−1.803 +
other support	3.328 (±1.081)	3.555 (±0.980)	−3.639 ***

Note: + *p* < 0.1, ** *p* < 0.01, *** *p* < 0.001; Health Risk Behavior Survey of Middle School Students (2015).

**Table 2 ijerph-18-03347-t002:** Violent experience and social support on NSSI and R-NSSI behaviors of male students.

Types of NSSI (Benchmark: N-NSSI)	Model 1		Model 2		Model 3	
	O-NSSI	R-NSSI	O-NSSI	R-NSSI	O-NSSI	R-NSSI
Violent experience						
verbal violence	1.075	0.871	1.067	0.809	0.967	0.799
physical violence	1.207	1.787 ***	1.256	1.865 ***	1.296	1.962 ***
visual violence	1.008	1.267 *	1.082	1.294 *	1.192	1.221
cold violence	1.088	1.662 ***	1.129 **	1.688 ***	1.335	2.155 ***
sexual violence	1.704 *	1.993 ***	1.991	2.374 ***	1.602	1.718 *
Social support						
family support			0.709	0.905	0.530 +	1.055
friend support			0.671	0.491 **	0.813	0.529 *
other support			1.046	1.172	1.338	1.239
Controlled variable						
Age					0.627 **	0.634 ***
Only-child					0.687	0.562 +
Parents’ relationship					1.081	0.842
Family financial status					0.889	1.124
Romantic relationship status					1.676	2.398 *
2 log likelihood	536.989		687.614		597.572	
Cox–Snell R^2^	0.14		0.217		0.265	
Nagelkerke R^2^	0.189		0.294		0.358	

Note: + *p* < 0.1, * *p* < 0.05, ** *p* < 0.01, *** *p* < 0.001; Health Risk Behavior Survey of Middle School Students (2015).

**Table 3 ijerph-18-03347-t003:** Violent experience and social support on NSSI and R-NSSI behaviors of female students.

Types of NSSI (Benchmark: N-NSSI)	Model 4		Model 5		Model 6	
	O-NSSI	R-NSSI	O-NSSI	R-NSSI	O-NSSI	R-NSSI
Violent experience						
verbal violence	1.083	1.446	1.014	1.510 *	1.073	1.489 *
physical violence	1.005	1.210	1.043	1.122	0.983	1.002
visual violence	1.271	1.655 ***	1.302	1.505 *	1.693 *	1.593 **
cold violence	1.071	1.070	1.057	1.096	0.931	1.036
sexual violence	1.787 *	1.083	1.357	1.055	1.844	1.192
Social support						
family support			0.713	1.150	0.721	1.253
friend support			0.823	0.578 *	0.803	0.546 *
other support			0.971	1.078	1.082	0.980
Controlled variable						
Age					0.623 ***	0.813 +
Only-child					0.450 +	0.995
Parents’ relationship					0.861	0.733 *
Family’s financial status					0.822	0.703
Romantic relationship status					1.254	1.661
2 log likelihood	346.600		593.281		486.974	
Cox–Snell R^2^	0.96		0.129		0.228	
Nagelkerke R^2^	0.123		0.166		0.292	

Note: + *p* < 0.1, * *p* < 0.05, ** *p* < 0.01, *** *p* < 0.001; Health Risk Behavior Survey of Middle School Students (2015).

## Data Availability

The data that support the findings of this study are available from the Institute for Population and Development Studies at Xi’an Jiaotong University, but ethical restrictions of Xi’an Jiaotong University apply to the availability of the data, which contain privacy variables that might affect the participants’ mental health. The data were used under license for the current study and, so, are not publicly available. Data are, however, available from the corresponding author upon reasonable request and with permission of the Institute for Population and Development Studies at Xi’an Jiaotong University.

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
