# Peer review of "Impact of Violent Experiences and Social Support on R-NSSI Behavior among Middle School Students in China"

_ijerph, 2021, doi:10.3390/ijerph18073347_

Round 1
Reviewer 1 Report
Abstract is well prepared and has all the information about the methodology and the main conclusions. It would have been interesting to mention the number of students involved in the study.
The keywords are appropriate. However, one of the keywords is “middle school students” and between lines 135 and 138 the term “Universities” is mentioned, which can lead to an unclear interpretation.
The introduction has a sufficient literature review on the study's keywords. Figure 1 clearly shows the main variables identified in the literature review and the relationship between them.
It would have been interesting to characterize the study population, namely regarding sex. How many students attend the schools under study? Why 1200 questionnaires? How were they distributed to schools?
The researchers state that the study was carried out at Universities (between lines 135 and 138) and in line 178 it is said that the average age of students is 14.63 years. Are these statements correct?
The statistical methods used in the investigation are adequate. For the use of multiple linear regressions, the authors do not mention whether the initial statistical assumptions were verified. In view of the sample size, more ambitious statistical methods could be used.
According to the results presented, discussion and conclusions are adequate.
Author Response
Dear reviewer,
Thank you for your valuable comments on our paper. We have revised our manuscript to incorporate your constructive comments and suggestions. The details of revisions are as follows.
Point 1: Abstract is well prepared and has all the information about the methodology and the main conclusions. It would have been interesting to mention the number of students involved in the study.
Response 1: According to your suggestions, we have mentioned the number of students involved in the study in page 1, line 15.
Point 2: The keywords are appropriate. However, one of the keywords is “middle school students” and between lines 135 and 138 the term “Universities” is mentioned, which can lead to an unclear interpretation.
Response 2: Thank you for the reminding.The schools mentioned in the article are all middle schools. Because in China, some middle schools are affiliated with some universities and so named after a secondary school affiliated with the university. By checking the information on the official website of the school, we have revised the details of the name of the investigated school. Please refer to the text for details in page 4, line 142-150.
Point 3: It would have been interesting to characterize the study population, namely regarding sex. How many students attend the schools under study? Why 1200 questionnaires? How were they distributed to schools?
Response 3: According to your suggestions, we have added the sex characterize of the study population in page 4, line 151-152. In addition, in the research, stratified proportional sampling method was adopted to select a total of 7 middle schools, including 4 of the key middle schools and 3 of the ordinary middle schools. Students of different grades and genders are selected on an equal basis in collaboration with liaison teachers from each school.Specific sampling plan is as follows: The High School Affiliated To Shaanxi Normal University 100, The Middle School Attached To Northwestern Polytechnical University 100, Xidian High School Attached To Xidian University 100, The High School Affiliated To Xi’an Jiaotong University 40 and Xi’anTie Yi High School 60, Taiyi Road Middle School 600 and Qinchuan Middle School 200. A total of 1,200 students participated finally.
Point 4: The researchers state that the study was carried out at Universities (between lines 135 and 138) and in line 178 it is said that the average age of students is 14.63 years. Are these statements correct?
Response 4: As explained in response 2, the study was carried out at middle schools, the average age of students is 14.63 years old is correct.
Point 5: The statistical methods used in the investigation are adequate. For the use of multiple linear regressions, the authors do not mention whether the initial statistical assumptions were verified. In view of the sample size, more ambitious statistical methods could be used.
Response 5: We agreed with you.According to your suggestions,the initial statistical assumptions were verified and the details were supplemented in page 5, line 200-201,page 6, line 274-276 and page 7, line 325-327..
Reviewer 2 Report
I am not able to comment on the submitted manuscript as extensive editing of English language and style required.
Author Response
We thank you for your thoughtful suggestions and insights. The manuscript has been rechecked and the necessary changes have been made. The responses to all comments have been prepared and attached given below.

Reviewer 3 Report
Some review remarks
29 - In 1. Introduction, I consider it important to provide a paragraph on the social and cultural context of the study.
128 - Figure 1. Problem behavior theory explanation model of R-NSSI behavior of adolescents
- Indicate a source
130 - 2. Materials and Methods:
- I think it would be important for the author to make the sample selection criteria presented should be clearer
198 - Table 1. Sex Differences in Variable Measurement:
- Indicate a source
272 - Table 2. Violent Experience and Social Support on NSSI and R-NSSI Behavior of Male Students:
- Indicate a source
321 - Table 3. Violent Experience and Social Support on NSSI and R-NSSI Behavior of Female Students:
- Indicate a source
492 - According to the nature of this study, does the author consider it relevant to put Educating for citizenship?
References:
Is Ok.
Final considerations:
Throughout the text it speaks of cold violence. This term needs clarification for a good international understanding.
The article presents scientific mastery at the level of methodological process and discussion of results It develops an innovative approach on the issues of violence among young people crossing behavioural analysis with differentiated social support.
The study leaves open an interesting line of research on the prevention of violent behaviour among young people, based on the promotion of social support that can be family, community, group and institutional.
It leaves open the field of study on collaborative methodologies in an integrated approach to social problems that affect people's lives.
Author Response
Dear reviewer,
Thank you for your valuable comments on our paper. We have revised our manuscript to incorporate your constructive comments and suggestions. The details of revisions are as follows.
Point 1: 29 - In 1. Introduction, I consider it important to provide a paragraph on the social and cultural context of the study.
Response 1:According to your suggestions, we have provided a paragraph on the social and cultural context about Chinese middle school students in introduction, in page 1-2, line 41-47.
Point 2: 128 - Figure 1. Problem behavior theory explanation model of R-NSSI behavior of adolescents
- Indicate a source
Response 2:Thank you for bringing this to our attention. The source of the Figure was not indicated, as the figure has been prepared by the authors themselves, and no third-part artwork was used. According to your suggestions, we have added a note in page 3, line 137.
Point 3: 130 - 2. Materials and Methods:
- I think it would be important for the author to make the sample selection criteria presented should be clearer
Response 3:According to your suggestions, we have explained the selection criteria of samples more accurately, including the specific sampling methods and sample objects in page 4, line 142-145.
Point 4: 198 - Table 1. Sex Differences in Variable Measurement:
- Indicate a source
Response 4:The data in table 1 came from the survey of “Health Risk Behavior Survey of Middle School Students” in 2015 by our research group, and the table was drawn by the authors. According to your suggestions, we have added the note in page 5, line 230-231.
Point 5: 272 - Table 2. Violent Experience and Social Support on NSSI and R-NSSI Behavior of Male Students:
- Indicate a source
Response 5:The data in table 1 came from the survey of “Health Risk Behavior Survey of Middle School Students” in 2015 by our research group, and the table was drawn by the authors. According to your suggestions, we have added the note in page 7, line 287-288.
Point 6: 321 - Table 3. Violent Experience and Social Support on NSSI and R-NSSI Behavior of Female Students:
- Indicate a source
Response 6:The data in table 1 came from the survey of “Health Risk Behavior Survey of Middle School Students” in 2015 by our research group, and the table was drawn by the authors. According to your suggestions, we have added the note in page 8, line 367.
Point 7: 492 - According to the nature of this study, does the author consider it relevant to put Educating for citizenship?
Response 7:Thank you for your advice. It's very important.In the conclusion part of the paper, we mentioned that we should further explore and innovate the new model of health education at home and school, and put forward the effective policy transformation(in page 11, line 515-520). However, our research at the current stage is limited, which is what we will further study in the follow-up research.
Point 8: Throughout the text it speaks of cold violence. This term needs clarification for a good international understanding.
Response 8:Thank you for bringing this to attention, we have added the definition of cold-violence on its first mention in the manuscript (page 2, line 58-60).